# The Risk of Recurrence in Endometrial Cancer Patients with Low-Volume Metastasis in the Sentinel Lymph Nodes: A Retrospective Multi-Institutional Study

**DOI:** 10.3390/cancers15072052

**Published:** 2023-03-30

**Authors:** Alessandro Buda, Cristiana Paniga, Salih Taskin, Michael Mueller, Ignacio Zapardiel, Francesco Fanfani, Andrea Puppo, Jvan Casarin, Andrea Papadia, Elena De Ponti, Tommaso Grassi, Jessica Mauro, Hasan Turan, Dogan Vatansever, Mete Gungor, Firat Ortag, Sara Imboden, Virginia Garcia-Pineda, Stefan Mohr, Franziska Siegenthaler, Stefania Perotto, Fabio Landoni, Fabio Ghezzi, Giovanni Scambia, Cagatay Taskiran, Robert Fruscio

**Affiliations:** 1Department of Medicine and Surgery, University of Milano-Bicocca, 20900 Monza, Italy; 2Clinic of Obstetrics and Gynecology, IRCCS San Gerardo, 20900 Monza, Italy; 3Division of Gynecologic Oncology, Ospedale Michele e Pietro Ferrero, 12060 Verduno, Italy; 4Department of Obstetrics and Gynecology, School of Medicine, Ankara University, 06620 Ankara, Turkey; 5Inselspital, University Hospital of Bern, University of Bern, 3010 Bern, Switzerland; 6Gynecologic Oncology Unit, La Paz University Hospital, 28046 Madrid, Spain; 7Department of Woman and Child Health and Public Health, Division of Gynecologic Oncology, Fondazione Policlinico Universitario A. Gemelli IRCCS, 00168 Rome, Italy; 8Department of Women and Child Health and Public Health, Università Cattolica del Sacro Cuore, 00168 Rome, Italy; 9Department of Obstetrics and Gynecology, Ospedale Santa Croce e Carle, 12100 Cuneo, Italy; 10Department of Obstetrics and Gynecology, ‘Filippo Del Ponte’ Hospital, University of Insubria, 21100 Varese, Italy; 11Department of Gynecology and Obstetrics, Ente Ospedaliero Cantonale, 6900 Lugano, Switzerland; 12Faculty of Biomedical Sciences, University of the Italian Switzerland, 6900 Lugano, Switzerland; 13Medical Physics Department, Foundation IRCCS San Gerardo Hospital, 20900 Monza, Italy; 14Department of Obstetrics and Gynecology, İstanbul Training and Research Hospital, University of Health Sciences, 34766 İstanbul, Turkey; 15Department of Obstetrics and Gynecology, School of Medicine, Koc University, 34450 İstanbul, Turkey; 16Department of Obstetrics and Gynecology, School of Medicine, Acibadem University, 34750 İstanbul, Turkey

**Keywords:** endometrial cancer, low-volume metastasis, sentinel lymph node biopsy, ultrastaging, recurrence-free survival

## Abstract

**Simple Summary:**

The surgical management of apparent early-stage endometrial cancer is still unclear. Nodal involvement is prognostic, but the role of retroperitoneal staging is still debated. Sentinel node mapping has been introduced and accepted as a valid alternative to full lymphadenectomy. Furthermore, ultrastaging provides a more accurate analysis of the excised lymph nodes by detecting a higher rate of low-volume metastasis. The aim of this study was to evaluate the impact of low-volume metastasis on recurrence-free survival in women with apparent early-stage endometrial cancer in a large retrospective multi-institutional collaboration.

**Abstract:**

The aim of this study was to assess the impact of low-volume metastasis (LVM) on disease-free survival (DFS) in women with apparent early-stage endometrial cancer (EC) who underwent sentinel lymph node (SLN) mapping. Patients with pre-operative early-stage EC were retrospectively collected from an international collaboration including 13 referring institutions. A total of 1428 patients were included in this analysis. One hundred and eighty-six patients (13%) had lymph node involvement. Fifty-nine percent of positive SLN exhibited micrometastases, 26.9% micrometastases, and 14% isolated tumor cells. Seventeen patients with positive lymph nodes did not receive any adjuvant therapy. At a median follow-up of 33.3 months, the disease had recurred in 114 women (8%). Patients with micrometastases in the lymph nodes had a worse prognosis of disease-free survival compared to patients with negative nodes or LVM. The rate of recurrence was significantly higher for women with micrometastases than those with low-volume metastases (HR = 2.61; *p* = 0.01). The administration of adjuvant treatment in patients with LVM, without uterine risk factors, remains a matter of debate and requires further evaluation.

## 1. Introduction

Endometrial cancer (EC) is the most common gynecological malignancy in developed countries, with 65,950 new cases predicted to occur in 2022 in the United States [1]. The rate of mortality has increased more rapidly than the incidence rate, despite the estimate that almost 67% of women are diagnosed with the disease confined to the uterine body [1,2]. Poor prognosis is associated with advanced-stage disease, high-risk histology such as serous carcinoma, age, grade, depth of myometrial invasion, lymph vascular space invasion, tumor size, and lower uterine segment invasion [3,4]. Surgical staging represents the main step in the treatment. Evaluation of lymphatic nodes status in endometrial cancer is recommended by many Gynecological Oncology Societies because it represents the most important prognostic factor, with implications in disease stage, prognosis, and guidance for adjuvant treatment. The last practice bulletin published by the ACOG committee in 2015, in association with the Society of Gynecologic Oncology, recommended retroperitoneal staging, reporting that “the initial management of endometrial cancer should include comprehensive surgical staging” [5]. However, randomized controlled trials failed to demonstrate the therapeutic role of lymphadenectomy in EC, reporting an increased risk of postoperative morbidity without significant impact on the oncologic outcomes [6,7]. Instead, sentinel lymph node (SLN) biopsy offers an interesting, intermediate compromise for patients with presumed early-stage endometrial cancer [8]. In recent years, several publications have shown that SLN mapping is an effective method for identifying disease in the lymph nodes, reporting high sensitivity, low false-negative rates, as well as good negative predictive values [9,10,11]. The most recent National Comprehensive Cancer Network (NCCN) and ESGO guidelines [12,13] support SLN mapping as part of the surgical staging of endometrial cancer, providing there is strict adherence to the principles and algorithm of the technique [14]. The surgical SLN algorithm has been proposed to reduce the false-negative rare in women with disease apparently confined to the uterine body. 

In the absence of SLN mapping on one or both sides, a side-specific lymphadenectomy should be performed, and any suspicious or enlarged node should be removed regardless of mapping [12,14]. Furthermore, surgeon expertise and attention to technical details are crucial. The SLN biopsy showed a decrease in morbidity compared to complete lymphadenectomy regarding lymphedema and lymphocele formation. Ultrastaging of SLNs showed a higher sensitivity in detecting nodal metastasis, mainly due to low-volume metastasis (LVM) detection. LVM could represent 25% to 63% of all positive SLNs, which means an 8% increase in determining nodal positivity compared to standard pathological staging. However, the therapeutic implications and prognostic value of LVM remain controversial. In addition, the current adjuvant management of these cases is still debated [15,16]. The aim of this study was to evaluate the low-volume metastasis impact on recurrence-free survival in surgically staged women with apparent early-stage endometrial cancer.

## 2. Materials and Methods

Data were retrospectively collected from 13 institutions across Europe. All patients underwent surgical staging, including total hysterectomy, bilateral salpingo-oophorectomy, and SLN mapping, with or without pelvic lymphadenectomy and aortic lymphadenectomy, between November 2012 and November 2020. Patients with sarcoma or metastatic disease at diagnosis were excluded. Informed consent was obtained from all subjects involved in the study. Ethical review and approval were waived for this study since data were properly anonymized and informed consent was obtained at the time of original data collection. All data were collected and stored in databases by each participating Center. The data required to create the database were extracted from patients’ medical records.

### 2.1. Sentinel Lymph Node Mapping Protocol

SLN assessment was carried out in all institutions following the Memorial Sloan Kettering algorithm [11]. Intraoperatively, 4 mL of patent blue (methylene blue or isosulfan blue), or indocyanine green (ICG), at a final concentration of 1.25 mg/mL, were injected into the cervix (2 mL for each side, 1 mL superficially and 1 mL deeply). At the beginning of the surgery, all the pelvic areas were carefully inspected, and SLNs were identified by following the colored lymphatic channels.

LVM was defined as isolated tumor cells (ITC: <0.2 mm largest diameter focus of metastatic disease per lymph node) or micrometastases (MM: between 0.2 mm and 2.0 mm focus of metastatic aggregate) in the SLNs. Patients with more than one metastasis, either in the same SLN or in multiple SLNs, were classified based on the diameter of the largest metastasis [14]. Women were divided into 2 subgroups: those with MM and/or ITC and those with macrometastases. Patients who completed nodal staging with pelvic ± aortic lymphadenectomy were also included upon the discovery of nodal metastasis in non-SLN.

### 2.2. Statistical Analysis

Qualitative variables were reported with absolute numbers and percentages. Quantitative variables were reported as median and range. Categorical variables were compared using the chi-square test for univariate analysis. A multivariate analysis was carried out using logistic regression. Disease-free survival (DFS) was calculated by using the Kaplan–Meier method and the Mantel–Cox statistical test. The alpha error was set at 5%. All statistical analyses were performed using the STATA software version 14.1 (Stata Corporation, College Station, TX, USA: StataCorp LP).

## 3. Results

A total of 1428 endometrial cancer patients were retrospectively collected from 13 centers. Table 1 presents the general characteristics of the study population.

The median age at surgery was 63 years (range 55–71), and the median BMI was 28 kg/m^2^ (range 24–33). SLNs were identified using the florescent dye Indocyanine Green (ICG) in 76.2% of the patients. Bilateral mapping was achieved in 78.8% of patients, whereas unilateral detection was achieved in 261 patients (18.3%). Overall, 186 patients (13.0%) were shown to have positive lymph nodes at final pathology. Only 19 (1.3%) women had non-SLN positive with negative SLNs. The median number of SLNs removed was 2 (range 2–4). Positive sentinel lymph nodes were mainly located in the external iliac (38.3%) and obturator (37.2%) regions. Among the 186 patients with positive LNs, 110 (59.1%) positive LNs included macrometastases, 50 (26.9%) micrometastases, and 26 (14.0%) ITC.

In patients with negative lymph nodes, adjuvant therapy was performed in 35.6%, whereas in low-volume metastasis, adjuvant therapy was administered in 84.2% of cases. Moreover, 94.5% of patients with lymph nodes macrometastases underwent adjuvant therapy (Table 2).

### Disease-Free Survival Data

At a median follow-up of 33.3 months, 298 women (90.9%) were alive with no evidence of disease. Forty-five women died of disease, and forty-nine (3.4%) are alive with disease. Overall, 114 patients (8%) relapsed: 5.9% of women with negative nodes (75), 28.2% of patients with macrometastases (31), and 14.5% of women with LVM (11).

Eight patients with macrometastases showed multiple sites of recurrence (26%), and six had a distant relapse (19%). Among patients with micrometastases and/or ITC, local pelvic, nodal, and multiple sites recurrences were 3 (27%), 3 (27%), and 3 (27%), respectively. A complete pelvic lymphadenectomy did not seem to affect the recurrence rate (*p* = 0.573). The 3-year DFS of women with negative lymph nodes, low-volume metastasis, and macrometastases was 90.6%, 84.3%, and 58.5% (*p* < 0.0001), respectively.

The Cox regression multivariate analysis showed that only the presence of LVSI was associated with recurrence. In addition, neither the type of nodal metastasis nor the administration of adjuvant therapy was statistically significant for the risk of recurrence (Table 3).

However, women with LVM showed a higher risk of recurrence when compared with patients with negative lymph nodes (HR = 2.34; *p* = 0.009). Moreover, in the subgroup of 185 women with positive lymph nodes, the 3-year recurrence-free survival was statistically significantly lower for women with macrometastases compared to patients with low-volume metastasis (HR = 2.61; *p* = 0.01).

## 4. Discussion

In this large retrospective multicenter study of patients with apparent early-stage endometrial cancer, we observed that women with nodal involvement had a worse prognosis of disease-free survival compared to patients with negative lymph nodes. Moreover, among women with nodal involvement, those with macrometastases showed a higher risk of recurrence compared to women with LVM, regardless of receiving adjuvant therapy. In patients with positive lymph nodes, the uterine risk factors significantly associated with recurrence were grade 3 disease and myometrial infiltration greater than 50%.

In the multivariate analysis, only the presence of LVSI was significantly associated with the risk of recurrence.

To date, in the absence of a large amount of prospective data, the evidence supports the treatment of women with micrometastases, whereas patients with isolated tumor cells with no uterine risk factors, such as endometrioid grade 1 disease and neither LVSI nor uterine serosal invasion, can be spared from adjuvant therapy [15,16].

A recent meta-analysis by Gòmez-Hidalgo et al. suggests a higher relative risk of recurrence in women with low-volume metastasis in the sentinel lymph nodes, independently from adding adjuvant therapy [17].

Similar to our results, Garcia-Pineda et al. [18] in their study included 230 patients that underwent nodal staging for early-stage endometrial cancer and reported a worsened progression-free survival for women with macrometastases compared to patients with isolated tumor cells and micrometastases in the SLNS (*p* < 0.05). In a retrospective study of 800 patients with endometrial cancer, Clair et al. [19] reported similar results for 3-year PFS for both negative node patients and low-volume metastasis (90% and 86%, respectively) and 71% for macrometastases (*p* < 0.001). The authors concluded that administering adjuvant therapy improves the prognosis of women with low-volume metastasis compared to patients with macrometastases.

Plante et al. also showed that the 3-year progression-free survival of women with ITC (95.5%) or micrometastases (85.5%) was similar to node-negative patients (87.6%). However, the result was statistically different when the LVM subgroup was compared to patients with macrometastases (58.5%). Although 68% of patients with isolated tumor cells received adjuvant treatment, the authors concluded that women with isolated tumor cells should be treated based on uterine risk factors [20].

A recent study by Backes et al. [21] reported that in 175 women with stage I–II endometrioid endometrial cancer, the presence of ITC in the SLNs did not affect the recurrence, regardless of the administration of adjuvant therapy. Similarly, Ghoniem et al., in a recent multi-institutional retrospective study including 247 patients with low-volume metastasis, showed that patients with ITC and grade 1 endometrioid disease without LVSI or serosal invasion had a favorable prognosis regardless of the administration of adjuvant therapy [16].

Sentinel node mapping has been adopted and accepted worldwide for nodal evaluation in women with low-grade endometrial cancer [12]. However, a recent review of the literature conducted by How et al. [22], they reported that SLN biopsy generated similar detection rates and accuracy as seen in low-grade disease. Although limited in retrospective study design and short-term follow-up, the studies included in that review have not demonstrated inferior survival outcomes of SLN mapping compared to traditional lymphadenectomy in high-grade endometrial cancer. Notwithstanding, the authors underlined the importance of obtaining sufficient operator experience with at least 30 cases and following the SLN surgical algorithm, which remains essential to preserving diagnostic accuracy.

In our study, the disease-free survival of the 79 women with LVM was not statistically different from those with negative nodes.

A recent retrospective study by Ignatov et al. [23] included 2392 patients with endometrial cancer with and without micrometastases. The authors showed that, with adjuvant therapy, the disease-free survival in the cohort of patients with micrometastases was reduced compared to disease-free survival in the node-negative cohort even after adjustment for age at diagnosis, myometrial invasion, histological grade, and the type and performance status.

To date, evidence from the majority of the studies suggests that adjuvant treatments in the presence of isolated tumor cells, any proposal of adjuvant treatments should be related to the presence of high-risk uterine factors. In the recent study of Bogani et al. [24], they included 572 patients who underwent hysterectomy with or without SLN biopsy. In the majority of cases, adjuvant therapy was administered based on the presence of uterine risk factors. The authors suggested that nodal evaluation might not be essential for tailoring the need for adjuvant therapy.

The strength of our study is inclusion of a large number of reviewed cases, the extensive experience with SLN staging, the high volume of cases at the participating institutions, and the similar approach to SLN mapping shared by all centers. However, the study has some drawbacks, including the retrospective design and the absence of similar adjuvant strategies amongst the different institutions, which might have influenced the results. Furthermore, the ultrastaging protocols for detecting low-volume metastasis vary among institutions globally; therefore, the SLN cannot be considered a standardized procedure in the absence of an international consensus for guidelines on an ultrastaging protocol [25]. As a result, the rates of nodal involvement seen in available studies in the literature are difficult to compare [26,27].

## 5. Conclusions

This study is one of the largest multi-institutional retrospective studies in women with apparent early-stage endometrial cancer.

We confirmed that macrometastases in the lymph nodes indicate a worse prognosis. However, patients with LVM also showed a higher risk of relapse compared to node-negative patients, regardless of the administration of adjuvant therapy.

The results of the ongoing SELECT prospective study, evaluating the impact of exclusive sentinel lymph node mapping on the intermediate risk of endometrioid endometrial cancer with no adjuvant therapy [28], and the ENDO-3 randomized trial [29] comparing simple hysterectomy alone with a hysterectomy and sentinel node mapping, are forthcoming. Our study confirms that women with positive nodes showed a higher risk of recurrence.

Furthermore, a higher risk was observed in women with macrometastases.

Integrating the molecular profile of high-risk endometrial cancer will probably provide new evidence to better help clinicians in choosing a more personalized adjuvant approach for every patient diagnosed with low-volume metastasis [30].

## Figures and Tables

**Figure 1 cancers-15-02052-f001:**
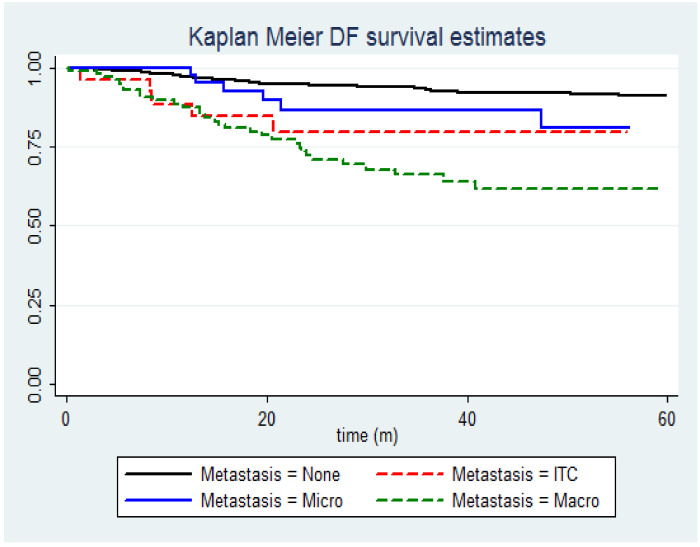
Kaplan–Meier plot of disease-free survival by the nodal status of lymph nodes. (N = 1428) Group 1 (black): negative; Group 2 (red dashed): isolated tumor cells; Group 3 (blue): micrometastases; Group 4 (green dashed): macrometastases.

**Table 1 cancers-15-02052-t001:** General characteristics of the study population (N = 1428).

Variables	No. Patients (%)
Age (at surgery), median (IQR)	63 (55–71)
BMI, median (IQR) Unknown	28 (24–33) 100
Histology Endometrioid Other histologies	1223 (85.6%) 205 (14.4%)
Grade Grade 1 Grade 2 Grade 3 Unknown	558 (39.1%) 541 (37.9%) 310 (21.7%) 19 (1.3%)
Myometrial infiltration <50% >50% Unknown	969 (67.9%) 456 (31.9%) 3 (0.2%)
Cervical stromal invasion NO YES Unknown	1268 (88.8%) 91 (6.4%) 69 (4.8%)
LVSI NO YES Unknown	1043 (73.0%) 310 (21.7%) 75 (5.3%)
FIGO stage final pathology IA IB II IIIA IIIB IIIC1 IIIC2 IVA IVB	877 (61.4%) 282 (19.8%) 60 (4.2%) 38 (2.7%) 6 (0.4%) 113 (7.9%) 46 (3.2%) 2 (0.1%) 4 (0.3%)
Surgical approach Minimally invasive surgery (MIS + Robotic) Open surgery MIS + open	1277 (89.4%) 144 (10.1%) 7 (0.5%)
Dye used ICG TC99 + blue Blue alone ICG + TC99 Other	1088 (76.2%) 72 (5.0%) 186 (13.0%) 79 (5.5%) 3 (0.2%)
Mapping Bilateral mapping Unilateral mapping No migration	1126 (78.8%) 261 (18.3%) 41 (2.9%)
Nr. SLN removed, median (IQR)	2 (2–4)
Pelvic LND NO YES	665 (46.6%) 763 (53.4%)
Aortic LND NO YES	1109 (77.7%) 319 (2.3%)
Tot patients SLN+ N° patients with only SLN+ N° patients with only Non-SLN+ N° patients with both SLN and non-SLN+	186 (13.0%) 107 (7.5 %) 19 (1.3%) 60 (4.2%)
Type of nodal metastasis SLN+ and non SLN+ ITC Micro-Mets Macro-Mets	26 (14.0%) 50 (26.9%) 110 (59.1%)
Adjuvant therapy NO YES Unknown	814 (57.0%) 603 (42.2%) 11 (0.8%)
Recurrence NO YES	1314 (92.0%) 114 (8.0%)

BMI = Body Mass Index; IQR = InterQuartile Range; LVSI = LymphoVascular Space Invasion; FIGO = International Federation of Gynecology and Obstetrics; MIS = Minimally Invasive Surgery; ICG = Indocyanine Green; TC99 = Tecnetium99; SLN = Sentinel Lymph Node; LND = Lymph Node Dissection; ITC = Isolated Tumor Cells.

**Table 2 cancers-15-02052-t002:** Baseline characteristics of patients with positive lymph nodes (LNs) (N = 186).

	ITC (N = 26)	Micro-Mets (N = 50)	Macro-Mets (N = 110)	*p*-Value
Histology Endometrioid Other histologies	19 (73.1%) 7 (26.9%)	41 (82.0%) 9 (18.0%)	78 (70.9%) 32 (29.1%)	0.353
Grade Grade 1 Grade 2 Grade 3 Unknown	8 (32.0%) 9 (36.0%) 8 (32.0%) 1	10 (20.4%) 31 (63.3%) 8 (16.3%) -	12 (10.9%) 37 (33.6%) 61 (55.5%) -	<0.0001
Myometrial infiltration <50% >50% Unknown	16 (61.5%) 10 (38.5%) -	16 (32.7%) 33 (67.4%) 1	25 (22.7%) 85 (77.3%) -	0.001
Cervical stromal invasion NO YES Unknown	24 (92.3%) 2 (7.7%) -	40 (83.3%) 8 (16.7%) 2	90 (82.6%) 19 (17.4%) 1	0.528
LVSI NO YES Unknown	8 (30.8%) 18 (69.2%) -	10 (20.8%) 38 (79.2%) 2	24 (21.8%) 86 (78.2%) -	0.593
Adjuvant therapy none EBRT +/− BRT RCT CT Other Unknown	7 (26.9%) 5 (19.2%) 12 (46.2%) 2 (7.7%) 0 (0.0%) -	5 (10.0%) 5 (10.0%) 25 (50.0%) 7 (14.0%) 8 (16.0%) -	5 (4.6%) 7 (6.4%) 78 (71.6%) 11 (10.1%) 8 (7.3%) 1	0.002
Recurrences NO YES	21 (80.8%) 5 (19.2%)	44 (88.0%) 6 (12.0%)	79 (71.8%) 31 (28.2%)	0.066

ITC = Isolated Tumor Cells; Mets = metastasis; LVSI = LymphoVascular Space Invasion; SLN = Sentinel Lymph Node; EBRT = external beam radiotherapy; RT = radiotherapy; RCT = chemoradiation; CT = chemotherapy.

**Table 3 cancers-15-02052-t003:** Univariate and multivariate analysis on the risk of recurrence.

	Univariate Analysis		Multivariate Analysis	
	OR (95% IC)	*p*-Value	OR (95% IC)	*p*-Value
Age at surgery	1.04 (1.02–1.06)	<0.0001	1.03 (1.00–1.08)	0.085
Grade	2.72 (2.08–357)	<0.0001	1.21 (0.64–2.28)	0.560
LVSI	5.50 (3.69–8.19)	<0.0001	7.93 (1.74–36.07)	0.007
Histology	3.43 (2.25–5.24)	<0.0001	2.16 (0.90–5.21)	0.086
Nodal status (ITC vs. Micro-Mets vs. Macro-Mets)	1.58 (0.93–2.68)	0.094	1.56 (0.86–2.83)	0.142
Adjuvant therapy	5.70 (3.58–9.06)	<0.0001	4.20 (0.46–38.19)	0.203

OR = odds ratio; IC = interval of confidence; LVSI = Lympho-Vascular Space Invasion; ITC = Isolated Tumor Cells; Mets = metastasis. The plots of the cumulative proportion of patients with disease-free survival showed that the recurrence rate was significantly higher for patients with macrometastases (HR = 2.51; *p* < 0.0001) compared to patients with LVM and negative nodes (Figure 1).

## Data Availability

Data are archived and available at the Medical Physics Foundation IRCCS San Gerardo Hospital, Monza, Italy. Elena De Ponti, froze the database and performed the data analysis.

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
