# Peer review of "The Risk of Recurrence in Endometrial Cancer Patients with Low-Volume Metastasis in the Sentinel Lymph Nodes: A Retrospective Multi-Institutional Study"

_cancers, 2023, doi:10.3390/cancers15072052_

Round 1

Reviewer 1 Report

I congratulate the authors on the study with a large sample of operated women (1428 patients treated). On the other hand, with 13 centers it is possible to collect a great deal of data. The study, although retrospective and on a large sample, confirms the current literature and does not deviate from the conclusions. What will be needed in the near future is a broad prospective assessment of LVM. Only in this way will we have certain and irrefutable data.

Author Response

Thank you for your comments. Ongoing trials will help to answer the issues still debate regarding sentinel node in endometrial cancer at early stage

Reviewer 2 Report

Large cohort study, but the final results might be influenced by extremely different surgical and adjuvant oncological protocols between institutions.

Author Response

Thank you for your comment. We agree with this observation on possible bias related to multi institutions' different adjuvant approaches. 

In the discussion this is already described (lines 252-254) underlining this study's limitations.

English was further revised as requested

Reviewer 3 Report

The present study shows the usefulness of quantifying the role of low-volume metastasis in SLN regarding patient outcomes and the possibility of their application in clinical practice. The authors provided adequate details on methodology, evaluation, findings, and investigations. The particularities and novelty of the article are well underlined in the results and conclusions sections. Given the bibliography, it is clear that the authors made a complete review of the literature beforehand. 

However, some suggestions in the discussion section could improve the quality of the article:

1-    What was the SLN detection rate in this study compared to the literature percentage of 92%-97%? How does the detection rate of SLN influence the recurrence rate in these cases?

2-    Evaluation of pre- and postoperative histological grading to increase the accuracy of the results.

3-    The preoperative histological appearance of grades 1-2 is insufficient to predict the risk of lymph node metastases. As a result, a preoperative screening risk model based on preoperative characteristics, imaging, and biomarkers should be developed.

4-    Implementing SLN mapping in many countries could be more complicated because low- and intermediate-risk EC patients are treated by general gynecologists and not by a gynecologic oncologist.

5-    The preoperative use of the molecular classification of endometrial cancer (POLE, TP53, MMRd mutations) could increase the accuracy of lymph node staging.

6-    Development of a robot-assisted augmented reality system for lymph node identification in laparoscopic gynecologic oncology surgery.

Kind regards

Author Response

1-    What was the SLN detection rate in this study compared to the literature percentage of 92%-97%? How does the detection rate of SLN influence the recurrence rate in these cases?

Response: as shown in table 1, the overall detection rate was 97%, with a rate of no migration of 2.9% and an optimal bilateral mapping of 78.8%.

We revised this data on the dataset and all the numbers were corrected. The results of high bilateral mapping reflect in our opinion the high level of all the selected highly trained centers that participated to the study.  However, the aim of our study was to evaluate survival data. The sensitivity and the accuracy of the SLN mapping technique have been already widely described in the available literature and were not the primary objective of the study.

2-    Evaluation of pre- and postoperative histological grading to increase the accuracy of the results.

Response: thank you for the observation. We could not retrieve all preoperative biopsies even considering the number of participating centers.
In addition, we do not believe that accuracy can be increased in light of the fact that pre-OP biopsy is often inadequate and the final pathological examination is often changed. Instead, the accuracy of preoperative work-up (imaging) remains crucial and allows sentinel node biopsy in almost all apparent early-stage cases.

3-    The preoperative histological appearance of grades 1-2 is insufficient to predict the risk of lymph node metastases. As a result, a preoperative screening risk model based on preoperative characteristics, imaging, and biomarkers should be developed.

Response: we fully approve of this comment. We're working hard to think about a preoperative algorithm to determine a risk model also including new molecular indicators if available in the near future.

4-    Implementing SLN mapping in many countries could be more complicated because low- and intermediate-risk EC patients are treated by general gynecologists and not by a gynecologic oncologist.

Response: Agree. This is a modern scenario, especially in developing countries.

In our opinion, international society should help small hospitals and developing countries to implement the treatment of women with malignancies in well-trained centers or help non-referee centers by applying a competency assessment tool for sentinel lymph node biopsy by minimally invasive surgery in endometrial cancer as already published (Moloney K, Janda M, Frumovitz M, et al, Int J Gynecol Cancer 2021).

5-    The preoperative use of the molecular classification of endometrial cancer (POLE, TP53, MMRd mutations) could increase the accuracy of lymph node staging.

Response: we agree. See the above response. In the future, the molecular characteristics of the tumor will further improve the accuracy of SLN mapping. Furthermore, as discussed will also be possible when it is indicated or not to perform retroperitoneal staging. The ENDO-3 prospective randomized trial has been designed with this intent.

6-    Development of a robot-assisted augmented reality system for lymph node identification in laparoscopic gynecologic oncology surgery.

Response: We also agree with this comment. Artificial intelligence and the implementation of robotic surgery will further improve sentinel node identification in all gynecological malignancies.

English revision was further performed throughout the paper.